# Neurologic Deficit Score at 4–5 Days Post-eCPR Predicts Long-Term Brain Dysfunction in Rats Following Cardiac Arrest

**DOI:** 10.3390/biom15050732

**Published:** 2025-05-16

**Authors:** Wolfgang Weihs, Alexandra-Maria Stommel, Andrea Müllebner, Alexander Franz Szinovatz, Matthias Müller, Ingrid Magnet, Michael Holzer, Andrey V. Kozlov, Sandra Högler, J. Catharina Duvigneau

**Affiliations:** 1Department of Emergency Medicine, Medical University of Vienna, 1090 Vienna, Austria; wolfgang.weihs@meduniwien.ac.at (W.W.); alexandra-maria.stommel@meduniwien.ac.at (A.-M.S.); alexander.szinovatz@meduniwien.ac.at (A.F.S.); matthias.mueller@meduniwien.ac.at (M.M.); ingrid.magnet@meduniwien.ac.at (I.M.); michael.holzer@meduniwien.ac.at (M.H.); 2Biological Sciences and Pathobiology, University of Veterinary Medicine Vienna, 1221 Vienna, Austria; andrea.muellebner@vetmeduni.ac.at (A.M.); sandra.hoegler@vetmeduni.ac.at (S.H.); 3Ludwig Boltzmann Institute for Traumatology, Research Center in Cooperation with AUVA, Austrian Cluster for Tissue Regeneration, 1200 Vienna, Austria; andrey.kozlov@trauma.lbg.ac.at

**Keywords:** rat model, ventricular fibrillation cardiac arrest, hippocampus, cortex, neurodegeneration, heme oxygenase, oxoglutarate dehydrogenase complex, enzyme activity

## Abstract

Cardiac arrest (CA) survivors often develop long-term neurological deficits, but its long-term impact on vulnerable brain regions and neurological outcomes remains unclear. In a previous CA model with conventional cardiopulmonary resuscitation, we found reduced heme oxygenase (HO) activity in the hippocampus and cortex 14 days post-CA, suggesting its potential as a functional outcome marker. Here, we used a rat model with 6 or 8 min of CA followed by extracorporeal cardiopulmonary resuscitation. While in the 6 min-CA group, 67% survived to day 14, increased mortality within 4 days resulted in only 33% survival in the 8 min group post-ROSC. All animals displayed neurological impairment assessed by daily neurologic deficit scoring (NDS). While deficits declined within the first 3–4 days in the 6 min-CA animals, the 8 min-CA group showed significantly worse neurological outcomes until day 14. Two weeks post-CA, neuroinflammatory and neurodegenerative markers (HO-1, TNF-R1, Iba1, and GFAP) were elevated in the hippocampus, while HO and 2-oxoglutarate dehydrogenase complex activities were reduced in all rats, indicating a decrease in anti-oxidative capacity and mitochondrial capacity for metabolizing glutamate. NDS at day 4–5 strongly correlated with the delayed CA-mediated enzymatic dysfunction determined in the hippocampus. This finding highlights this time point for identifying at-risk individuals and suggests a prolonged therapeutic intervention lasting at least until 4 days post-CA.

## 1. Introduction

Post-cardiac arrest (CA) hypoxic–ischemic brain injury is a major component of post-CA syndrome, influenced by arrest duration and cause. Prolonged CA and resuscitation can lead to death or persistent neurological deficits [1], driven by excitotoxicity [2], oxidative stress, inflammation, and apoptosis [3]. The hippocampus (Hc), essential for memory and cognition [4], is particularly vulnerable [5] due to its high metabolic demand and limited oxygen reserves [6,7]. Despite advances in resuscitation, effective neuroprotective strategies remain lacking.

While the immediate impact of CA on brain function is well-documented, long-term effects on vulnerable regions like the Hc, cortex, and striatum are less understood. Studies, including our own [8,9], have demonstrated structural and functional changes in these areas that associate with sustained neurological deficits [10].

Conventional cardiopulmonary resuscitation (CPR) is the primary treatment to achieve return of spontaneous circulation (ROSC), but extracorporeal CPR (eCPR) is now recommended for selected patients unresponsive to standard measures [11]. Neurological prognostication after conventional CPR begins on day 3 [1], yet eCPR patients require a more tailored approach [12], due to prolonged recovery, with no established prediction model available.

Few long-term survival studies in small-animal models use eCPR after ventricular fibrillation CA (VFCA). Reproducible VFCA-eCPR models are crucial for studying pathophysiological mechanisms and potential therapies [13,14]. Our research suggests that an eCPR rat model provides superior standardization over manual or mechanical chest compressions [13], enabling investigation of delayed brain changes post-CA.

Neuroinflammation, reactive oxygen species (ROS) generation, mitochondrial dysfunction, and glutamate toxicity drive neuronal damage after CA. Oxidative stress and sustained extracellular glutamate elevation exacerbate excitotoxicity [15], perpetuating neuronal injury [16]. Two key enzyme systems—heme oxygenase (HO) and the oxoglutarate dehydrogenase complex (OGDHC)—are implicated in these processes. Two isoforms of HO, the constitutive HO-2 and the inducible HO-1, work in concert to maintain brain homeostasis. Both isoforms degrade heme into biliverdin, CO, and free iron, but they serve distinct roles in neuronal function. HO-2, the predominant isoform in neurons, is involved in regular heme turnover and the production of CO, which acts as a modulator of cerebral blood flow and neurotransmitters [17]. Induction of HO-1 mediates protection in response to acute stressful conditions, predominantly via the generated products, which can be mimicked by the application of HO products themselves. Biliverdin has important anti-inflammatory and antioxidant properties [18], and its administration was shown to reduce cerebral ischemia–reperfusion injury in rats [19]. Application of CO during extracorporeal resuscitation reduced neurologic damage after CA [20]. However, prolonged HO-1 activation may potentially cause harm via the increased generation of CO, which may contribute to ischemia-triggered neuroinflammation involving activation of cyclooxygenase-2 [21].

Our previous studies demonstrated that CA, despite increasing HO-1 expression, decreased HO-2 protein and HO activity in the Hc and motor cortex (mC), which correlated with increased neuroinflammation [9]. Given HO’s role in counteracting glutamate-induced cytotoxicity [22], its dysregulation may significantly contribute to neuronal damage seen in vulnerable brain regions. OGDHC controls the rate of the entire tricarboxylic acid (TCA) cycle and is crucial for energy metabolism. It links carbohydrate and amino acid metabolism and controls glutamate levels [23], thereby preventing its toxic accumulation. Oxidative stress, particularly lipid peroxidation products, inhibit OGDHC [24,25], resulting in compromised energy metabolism, increased ROS production, and apoptotic cell death [23,26]. Many different neurodegenerative processes are associated with OGDHC dysfunction [26,27]. We previously demonstrated that traumatic brain injury results in decreased OGDHC activity and neurotoxicity [28], and showed that supplementation with thiamine, a necessary cofactor, restores enzyme function [29].

This study examined whether delayed neurodegeneration, occurring two weeks post-CA, is linked to changes in HO and OGDHC activity in the Hc and mC. We aimed to determine if early neurologic deficit scores predict long-term enzymatic dysfunction, providing potential biomarkers for guiding neuroprotective strategies.

Research Question:

Does the severity of neurological deficits within the first week post-CA correlate with HO and OGDHC activity in the Hc and cortex two weeks later? Additionally, which neurologic deficit scoring (NDS) time point best predicts long-term enzymatic alterations and sustained neuronal dysfunction?

## 2. Materials and Methods

The experimental protocol was approved by the Intramural Committee for Animal Experimentation of the Medical University of Vienna and the Austrian Federal Ministry of Education, Science and Research (GZ: BMBWF-66.009/0315-V/3b/2018) and is visualized in the following scheme (Figure 1). The experiments were conducted in compliance with the European Union Directive 2010/63/EU and the ARRIVE guidelines [30].

### 2.1. Animal Model of VFCA and eCPR

The details of the experimental VFCA (6 and 8 min) and eCPR approach and the associated survival rates, as well as the manifestation of histological changes in the hearts and Hcs of the surviving rats, were recently described [31]. To answer our research question, 40 male Sprague–Dawley rats, aged 10–12 weeks and with a 450–600 g body weight (BW), obtained from the Division of Laboratory Animal Science and Genetics, Himberg, Austria, were used. Prior to initiation of the experiments, the animals were allowed to acclimatize for 14 days at the Center of Biomedical Research and Translational Surgery of the Medical University of Vienna. The animals were randomly allocated into two resuscitation groups and subjected to either 6 min (n = 20) or 8 min of CA (n = 25) followed by eCPR [31]. In this study, only those rats that achieved sustained ROSC were included. Additionally, a sham-operated group (n = 10) was included (Figure 1A).

Briefly, the rats were anesthetized via inhalation with 6% sevoflurane (FiO_2_ 1.0) and received subcutaneous piritramide (3 mg/kg BW) for analgesia. Core body temperature was maintained at 37 ± 0.2 °C using a heated operating table, and anesthesia was continued with 3.5% sevoflurane. Following intubation, mechanical ventilation was initiated with a respiratory rate of 65 breaths/min and a tidal volume of 7 mL/kg BW, using an FiO_2_ of 0.3. Physiological parameters, including electrocardiograms, esophageal and rectal temperatures, arterial blood pressure, end-tidal CO_2_ (etCO_2_), and peripheral oxygen saturation (SpO_2_), were continuously monitored.

VF was induced by delivering a 12 V/50 Hz alternating electrical current, with a maximum of 8 mA, via a pacing catheter placed next to the right heart within the central venous system, as detailed previously [8]. After removing the pacing catheter, the extracorporeal membrane oxygenation (ECMO) system was connected to the venous drainage cannula. Via open reservoir, heparin (200 IU/kg BW), epinephrine (20 μg/kg BW), and sodium bicarbonate (1 mmol/kg BW) were then introduced into the ECMO’s priming solution (ELO-MEL). Resuscitation was started at 6 or 8 min after the onset of CA with the initiation of ventilation and eCPR (100% oxygen at a gas flow of 100 mL/min). The ECMO flow started with 30 mL/min/kg BW and lasted until 100 mL/min/kg BW was reached. Two minutes into eCPR, two five-joule biphasic shocks were delivered every two minutes. Mechanical ventilation continued at a rate of 20 breaths/min with an FiO_2_ of 1.0, and epinephrine (10 μg/kg BW) was administered intravenously at 2 min intervals throughout the eCPR until ROSC was achieved within a maximum of 10 min. Following the achievement of ROSC, ECMO support was discontinued, and ventilation parameters were adjusted to a respiratory rate of 65 breaths per minute with a fraction of inspired oxygen (FiO_2_) of 0.5. The venous ECMO cannula was promptly removed post-ROSC, and the insertion site was aseptically closed. In each group (6 min-CA and 8 min-CA), 15 animals displayed “sustained ROSC”, characterized by maintenance of a mean arterial pressure (MAP) above 60 mmHg for at least 20 min without mechanical support, and were thus enrolled in this study. Arterial blood gas analyses were performed at 5, 15, and 60 min after ROSC. Continuous monitoring was maintained until the catheters were removed, followed by weaning and extubation. In the first 24–48 h post-ROSC, the rats typically remained unconscious and were housed individually until full recovery. Analgesia was administered using piritramide (3 mg/kg BW s.c.) every 6 h if signs of pain were evident. Supplemental fluids, NaCl 0.9% and glucose 10%, were provided s.c. if the rats were unable to drink. Unconscious rats were placed on a heating pad with a control unit linked to a rectal temperature probe to maintain normothermia. Animals requiring extended care received checks every 6–8 h, including pain-relief administration, fluid therapy, and assistance with hand- or tube-feeding. From the animals subjected to 6 min-CA, 10 survived until day 14, while only 5 animals survived from the group subjected to 8 min-CA. In the sham group, which underwent the same surgical procedures without the induction of VF and consequent CA and received identical care and medications, all animals survived until day 14.

### 2.2. Neurologic Deficit Scoring

The neurologic outcomes of the animals were assessed daily using neurologic deficit scoring (NDS] [32]. NDS consists of five different components: general deficits, cranial nerve reflexes deficits, motor deficits, sensory deficits, and coordination deficits, and quantifies neurological impairment on a scale from 0–10% (normal) to 100% (brain death). For the motor skills test, each limb was examined for its capacity for physiological movement. If this was not possible, 2.5 points were given per limb, up to a maximum of 10. The remaining points for motor skills resulted from the points for the travel ledge, placing test (front paws reaching when lifted from the ground by the tail), and righting reflex (attempting to right itself when placed on its back). The sensory system of the extremities was tested by gentle stimulation with a needle, the response to which should be retraction of the limb or twitching of the muscles. NDS was reported for day 1 to day 6 for animals that survived until the study’s endpoint 14 days post-CA. The neurological scoring system for NDS is listed in Table 1.

### 2.3. Sampling

This study analyzed data and tissues from animals that survived to the planned endpoint, allowing for immediate tissue collection and chilled tissue processing after euthanasia. Thus, animals that died before the endpoint were excluded. Histological and biochemical analyses were performed by blinded investigators. Two weeks post-resuscitation, the surviving animals were deeply anesthetized using sevoflurane and piritramide and subsequently euthanized. For each animal, the brain was then carefully extracted from the skull. For histological analysis, one randomly selected hemisphere of the brain was preserved in a 4% buffered formaldehyde solution. For other analyses, the mC and Hc were sampled from the second hemisphere. The mC was taken from a coronary section rostral to Bregma 1.7, and the Hc was taken from a coronary section caudal to Bregma −5.2 [33]. The sampling was performed on ice within 30 min. Brain regions were snap-frozen in liquid nitrogen and stored at −80 °C until use. The sampling process with the analyses performed is visualized in the scheme (Figure 1B).

### 2.4. Histological Analysis of Brain Regions

For histological evaluation, fixed brain halves were cut into coronary sections and the hippocampus and frontal cortex were embedded in paraffin. Sections of 2 µm thickness were cut and stained with hematoxylin and eosin (HE). For the detection of protein expression, immunohistochemistry (IHC) was performed using primary antibodies against HO-2, OGDH, glial fibrillary acidic protein (GFAP), and ionized calcium-binding adapter molecule 1 (Iba1). Details regarding antibodies, dilution, pretreatment, and detection systems are presented in Appendix A.

Hc and mC tissues were evaluated descriptively in HE-stained sections. In neurons of the hippocampal CA1 region and the motor cortex, protein expression of HO-2 and OGDH was rated semi-quantitatively in IHC-stained sections. Furthermore, HO expression in glial cells was assessed semi-quantitatively in the hippocampal CA1 region. Activation of astrocytes and microglia was rated similarly in GFAP- and Iba1-stained sections of mCs. For the semi-quantitative scoring, protein expression was assessed in four classes, physiologic (0), mild (1), moderate (2), and severe (3) increases in staining intensity in the respective brain regions. If necessary, intermediate steps were used. For white balance and assembly of representative histological images, Adobe photoshop CC 2025 was used.

### 2.5. Preparation of Tissue Homogenates

In order to avoid biases, which may have affected the experimental results due to inhomogeneity of the tissue pieces, we prepared homogenates of Hc and mC, which were used simultaneously for the determination of mRNA levels, NOx, and the enzyme activities (Figure 1). Frozen tissue pieces were weighed and transferred frozen into an Elvehjem potter. The tissue was supplemented 1:10 (*w*/*v*) with homogenization buffer (300 mmol/L sucrose, 20 mmol/L Tris, 2 mmol/L EDTA; pH 7.4) and homogenized with PTFE pestle on ice. The homogenates were distributed into 100 and 200 µL portions, which were immediately frozen in liquid nitrogen. The aliquots were stored at −80 °C until use. The protein concentration in each homogenate was controlled using the Bradford protein assay. For a given brain region, homogenates of the CA animals and the respective sham animals showed no differences in protein concentrations. Thus, we could exclude an effect of CA or sample preparation on the protein contents of the analyzed brain regions, justifying the parallel analysis of enzyme activities (Section 2.5. and Section 2.6) and levels of oxidation products of NO (NOx, Section 2.7), as well as gene expression levels (Section 2.8).

### 2.6. Determination of Heme Oxygenase Activity in Brain Regions

The determination of enzyme activities (HO and biliverdin reductase (BVR)) was performed as follows. An aliquot of freshly unfrozen homogenate (50 µL for the determination of HO activity, corresponding to approximately 0.5 mg of protein, and 25 µL for the determination of BVR activity) was added to a reaction mixture containing 500 nmol NADPH (Sigma-Aldrich, Germany) in a total volume of 150 µL assay buffer (100 mmol/L potassium phosphate buffer, 1 mmol/L EDTA; pH 7.4) supplemented with 20 nmol of hemin (for determination of HO activity) or with 200 nmol BV (for determination of BVR) and 250 nmol of NADPH in duplicates. The mixture was incubated under constant agitation in darkness for 30 min at 37 °C. The reaction was stopped by transferring the samples onto ice. BR was extracted into 1 mL of benzene. BR concentration was determined using a double-beam spectrophotometer (U-3000, Hitachi) and a standard calibration curve, which was generated by adding known amounts of BR to assay buffer followed by subsequent extraction. The detection limit of BR using this method was determined as 3 pmol BR. In all tissues, the BVR activity (the capacity to convert BV into BR) was much higher (i.e., 10 times) than that of HO (the capacity to convert heme to BR). This indicates that BVR activity is not rate-limiting and that all BV formed by the HO enzyme is completely reduced to BR by the underlying BVR. Enzyme activities were expressed as pmol BR formed per mg brain tissue in 30 min.

### 2.7. Determination of Oxoglutarate Dehydrogenase Complex Activity in Brain Regions

Determination of the rate-limiting activity of the TCA cycle enzyme OGDHC was performed as follows. Homogenates were sonicated in an ultrasonication bath (Thermo Fisher Scientific, CA) for 30 s at maximum power (1.5 kJ); sonication was repeated 3 times with 30 s gaps. The enzyme solubilization was completed by adding one volume of 4-fold concentrated RIPA buffer, that is, 40 mmol/L Tris-HCl buffer, pH 7.4, including 600 mmol/L NaCl, 4 mM EDTA, 1% sodium deoxycholate, and 4% NP-40, to the three volumes of the sonicated homogenate. The reaction mix consisted of 200 μL incubation buffer (1 mmol/L calcium chloride, 0.05 mmol/L coenzyme A, 1 mmol/L dithiothreitol (DTT), 1 mmol/L magnesium chloride hydrate, 50 mmol/L MOPS, 2.5 mmol/L NAD+; pH adjusted to 7.2), 50 μL homogenate, 2.8 mmol/L 2-oxoglutarate, and 0.62 mmol/L TPP. All reagents were obtained from Sigma-Aldrich, Germany. Measurements were conducted on a black 96-well culture plate (Sigma-Aldrich, Germany), with kinetic monitoring of fluorescence of the reaction product NADH at 460 nm, with an excitation wavelength of 340 nm.

### 2.8. Determination of Nitric Oxides (NOx)

Concentrations of NOx (NO_2_^−^ + NO_3_^−^ + SNO^−^) in the samples were determined using a chemiluminescence-based assay on the Sievers 280i Nitric Oxide Analyzer (General Electrics, Boulder, CO). Prior to analysis, the trap chamber was filled with 1 mol/L NaOH (Sigma-Aldrich, Steinheim am Albuch, Germany) and the reflux chamber was filled with the reduction agent (0.8% vanadium (III) chloride (Sigma-Aldrich) in 1 mol/L HCl (Sigma-Aldrich)). The latter was used to reduce all NOx species to nitric oxide. The temperature of the reaction chamber was set to 95 °C. To quantify the NOx levels, sodium nitrite standards were used, prepared as aqueous solution of NaNO_2_ (Sigma-Aldrich) in a concentration range of 10 nmol/L–10 µmol/L.

### 2.9. Analysis of Gene Expression

Gene expression analysis was performed using qPCR in accordance with Kozlov et al. [34]. Briefly, RNA was isolated from 100 µL frozen tissue homogenate using 1 mL of TriReagent™, in accordance with the manufacturer’s protocol. The quantity and purity of the extracted RNA were assessed by measuring absorbance at 260 nm and the ratio of absorbance at 260/280 nm using a Spark microplate reader equipped with a NanoQuant plate (Tecan Trading AG, Switzerland). We reverse transcribed 1 µg of RNA to copy DNA using an anchored oligo-dT primer and SuperScript II reverse transcriptase (Thermo Fisher Scientific, CA) in accordance with Mkrtchyan et al. [29]. By pooling equal aliquots from each cDNA, we generated an internal standard (IS), which we used as a reference for the quantification by qPCR. The primer pairs used for analyzing the mRNA levels of HO-1, HO-2, TNF-R1, the subunits of the OGDHC, 2-oxoglutarate dehydrogenase (OGDH, synonymous with E1), and dihydrolipoyl dehydrogenase (DLD, E3), as well as the internal reference genes hypoxanthine ribosyltransferase (HPRT) and cyclophilin A (Cyc), are described in detail in Appendix A. We performed the qPCR reaction with an iTaq DNA polymerase (Bio-Rad, Hercules, CA, USA) on a CFX96™ thermocycler (Bio-Rad, Hercules, CA, USA). We analyzed the data with the inbuilt software CFX manager (version 3.0, Bio-Rad) in the linear regression mode. We calculated target gene expression against IS using a modified ΔΔCq method and normalized for the relative expression values obtained for the internal reference genes HPRT and Cyc, as previously described [35]. Results are presented as fold changes relative to the means of the sham controls.

### 2.10. Data Analysis and Statistics

Data visualization and analysis were carried out using MS Excel, MS powerpoint, GraphPad Prism (software version 10.0.1 for Windows), SPSS (version 28; SPSS, IBM Corporation, Somers, NY, USA). Kaplan-Meier analysis was performed in R version 4.4.0 (https://www.R-project.org/ accessed on 8 May 2025) using packages readxl version 1.4.3 (https://CRAN.R-project.org/package=readxl accessed on 8 May 2025), survival version 3.6-4 (https://CRAN.R-project.org/package=survival accessed on 8 May 2025), and survminer version 0.5.0 (https://CRAN.R-project.org/package=survminer accessed on 8 May 2025). Group values were averaged and displayed as bars indicating SEMs; additionally, single data for each animal were provided. Cumulative survival analysis was visualized using Kaplan–Meier plots. Data were controlled for normal distribution using the Kolmogorov–Smirnov test. Data for which the hypothesis was not rejected were analyzed for group differences using the Student’s *t*-test (metric data). For data with a non-parametric distribution, the Kruskal–Wallis test was applied, followed by Dunn–Bonferroni correction. Paired analyses for non-parametric data were performed using the two-factor Friedmann test, corrected by the Bonferroni method. The significance level was set to 0.05. Analysis for significant correlations of parameters was performed using Spearman’s Rho and graphically displayed by a correlation matrix. Details of the statistical analyses are given in the legends to each figure.

## 3. Results

### 3.1. Animal Survival and Outcome

In the 6 min-CA group, out of all the 15 rats that achieved ROSC, 5 died within the first 2 days after ROSC and 10 (67%) survived until day 14 (453 ± 22 g at base line, 447 ± 29 g at sacrifice).

Out of the 15 animals that achieved ROSC in the 8 min-CA group, 10 died subsequently during the first 4 days after ROSC and only 5 (33%) survived until planned scarification (565 ± 46 g at BL; 534 ± 78 g at sacrifice) (see Figure 2). Values of blood gas analyses did not differ between the 6 min-CA and 8 min-CA groups. No animals died after day 4 in either group. We therefore assume that the transition from an acute, life-threatening phase caused by the CA–resuscitation- and acute ischemia–reperfusion-mediated complications to a recovery phase may take up to 4 days, i.e., 120 h, in our model.

### 3.2. Neurologic Deficit Scores Within the First Week After CA

Daily scoring of the neurological deficits (as is listed in Table 1) for the 14-day survivors showed that animals in the 8 min-CA group were clearly more compromised throughout the entire observation period (Figure 3). Sham animals displayed an NDS of 0 (no impairment) throughout the entire observation period. Neurological deficits declined throughout the initial days, indicating gradual recovery from the acute insult. Compared to the 6 min-CA group, the 8 min-CA group displayed significantly higher NDS values at 24, 96, and 120 h, highlighting the stronger and sustained neuronal impairment. Additionally, the 6 min-CA group recovered faster, which is demonstrated by the significantly lower NDS values obtained at days 3 to 6 compared to the ones obtained at 24 h and those obtained at days 5 and 6 when compared to the ones obtained at day 2 (48 h). In contrast, NDSs in the 8 min-CA group were significantly lower only at days 5 and 6 when compared to the 24 h values and did not significantly change after day 3. In both groups, despite improvement, some animals did not recover until 144 h after CA, suggesting delayed neuronal dysfunction.

### 3.3. Determination of Neuroinflammation and Neurodegeneration in the Hippocampus, but Not in Cortex Tissue, 14 Days After CA

In HE-stained sections of animals subjected to CA, mild to severe lesions were present in the CA1 region of the Hc. Pyramidal neurons were absent or showed degenerative changes, whereas glial cells showed an activated morphology. Microglial nuclei were elongated and oriented with their longitudinal axes perpendicular to the pyramidal layer. Astrocytes showed a gemistocytic morphology with increased cytoplasm and large light-blue nuclei. Markers indicating activation of astrocytes/microglial cells (GFAP and Iba1) showed increased levels in the CA1 region of the Hc of CA animals (see Appendix A), as was previously described in this model [31]. Analyses by qPCR confirmed immunohistochemical findings in this brain region. Levels of HO-1 mRNA, indicative of elevated cell stress sensed by glial cells [36], and TNF-R1 mRNA levels, indicative of neuroinflammatory processes, were significantly increased in Hc tissue of animals subjected to CA (see Figure 4A,B). Both markers tended to be higher in the 8 min-CA group compared to the 6 min-CA group. Due to the low numbers of animals in the latter group, differences did not reach significance (Figure 4). In contrast to Hcs, the mCs of animals subjected to VFCA did not show histological lesions in HE staining. Neither markers for cell stress (HO-1; Figure 4A), neuroinflammation (TNF-R1; Figure 4B), nor those indicating astrocyte/microglia activation (GFAP/Iba1; see Appendix A) revealed significant differences between the CA and sham animals.

Since these findings corresponded to the results of our previous study [9], we next examined HO activity in the Hc and mC. Additionally, we further determined the activity of the mitochondrial enzyme complex OGDHC. Both of these enzyme systems are essential for supporting neuronal function.

### 3.4. Determination of Enzyme Activities Relevant to Neuronal Function (HO and OGDHC) in Homogenates of the Hippocampus and Cortex 14 Days After ROSC

HO activity was significantly lower in the Hcs of animals in the 8 min-CA group (Figure 5A).

Although not significant, HO activity tended to decrease in the 6 min-CA group (Figure 5A). Thus, our data obtained for the Hc confirm the previously described results [9,13,37] and demonstrate that long-term derangements in this brain region following experimental CA occur rather independently of the applied resuscitation regime, namely, conventional CPR or extracorporeal CPR (eCPR). HO activity in the mC, contrary to our previous study, did not change, although the 8 min-CA group tended to have decreased activity (Figure 5A). Neither expression of BVRA nor BVR, the secondary enzyme of the heme degradation pathway, was significantly affected in either region.

OGDHC displayed significantly lower activity in the Hcs of rats subjected to CA two weeks after ROSC, which was additionally dependent on the CA duration. Albeit not significantly, OGDHC activity was also lower in mC tissue in the 6 min-CA group, similar to the situation for HO (Figure 5B). Of note, the activities of both enzyme systems correlated significantly with each other (Table 2, Figure 8) within and among tissues (Table 2). Expression levels for mRNA of the OGDHC subunits, E1 (OGDH) and DLD (E3), were not affected in either region in rats subjected to CA.

### 3.5. Determination of Oxidation Products of NO and Expression of iNOS

Since reactive oxygen and nitrogen species (RONS), especially peroxynitrite, are potent inhibitors of the catalytic activity of HO [38] and OGDHC [39,40], we questioned whether increased levels of oxidative stress metabolites could account for the decreased enzyme activity which we determined in the hippocampus. Contrary to our expectation, the levels of NOx were not increased but even lower in the Hc. Additionally, NOx-levels showed a dependence on the duration of CA, resulting in significantly lower levels in the Hcs of the 8 min-CA animals (Figure 6A). We found no changes in the mCs. Thus, RONS could be excluded as causative agents for the decreased enzyme activities. This interpretation is in line with the unchanged levels of iNOS mRNA (Figure 6B), the predominant enzyme isoform that is responsible for increased NO production, which is required for RONS generation. Since NO is also an important neurotransmitter for neurons, we next questioned whether the decreased NO levels which we determined in the Hc would result from a loss of neurons in that region.

### 3.6. Semi-Quantitative Determination of HO-2- and OGDH-Positive Cells in Hippocampus and Cortex Tissue by Immunohistochemistry 14 Days After ROSC

The IHC analysis revealed significantly decreased HO-2-specific staining of hippocampal neurons in the 6 min- and 8 min-CA animals compared to the sham animals. In contrast, glial cells in the CA1 regions of animals subjected to CA showed significantly increased HO-2 protein expression compared to the sham animals. However, no differences were detectable between the two CA durations. In the cortex, HO-2-specific staining of neurons was not significantly different between the groups (Figure 7A,C). OGDH-specific staining was lower in the Hc, independently of the duration of CA. In cortex tissue, we found a decreased intensity of OGDH-specific staining in neurons, which was dependent on the CA duration (Figure 7B,D).

### 3.7. Correlation Between Parameters Determined in the Brain Regions, Hippocampus and Motor Cortex, of Rats Two Weeks After CA and Their Association with Early NDS

Although changes in all parameters determined in the brain at 2 weeks after CA were moderate in the mC compared to the Hc, we found a strong correlation of the different parameters between each other in both tissues when analyzing all animals subjected to CA together (Figure 8). We found the strongest correlations between OGDHC and HO activity in both regions. The decrease in enzyme activity, particularly that of OGDHC, correlated inversely with the mRNA expression of DLD, one subunit of OGDHC, indicating an attempt to compensate for the decrease in activity within the Hc (Figure 8). This finding is in strong contrast to the positive correlation of HO-2 and OGDH proteins present in neurons with mRNA levels for HO-2 and E1, indicating a close relation of these markers. Further, an increased HO-1 correlated with a decreased OGDH presence in neurons in the Hc, suggestive of increased cell stress, when OGDH was low. Besides these tissue-specific correlations of parameters determined within brain tissue at 2 weeks after CA, we found a strong association with early NDS—a strong inverse correlation was found with enzyme activities of HO and OGDHC at 96 and 120 h after CA (Figure 8, Table 3, Figure 9).

### 3.8. Association of Early Neurologic Deficit Scoring with the Manifestation of Delayed Changes in Enzyme Activities in Hippocampus and Cortex Tissue 14 Days After CA

In order to understand the suitability of NDS at early time points for predicting neuronal dysfunction, we investigated how NDS determined within the first week after CA associated with molecular markers determined two weeks after CA in both brain regions. Immediate NDS (24 h) showed a positive correlation with the molecular stress marker HO-1 at the mRNA level in the cortex two weeks later (Table 3), reflecting the acute intensity of stress experienced during CA-mediated ischemia–reperfusion brain injury. However, no other functional parameter, which stands for the long-term neurological outcome, correlated with the 24 h NDS (Figure 8). Instead, the activity of HO and OGDHC, essential enzyme systems relevant for neuronal function, showed highly significant inverse correlations with the NDS recorded only at later time points, days 4 and 5.

Thus, NDS determined in animals subjected to CA that survive the critical 4 days after ROSC is predictive for an impaired long-term neurological outcome.

## 4. Discussion

Our study aimed to investigate whether the progression of neurological dysfunction in animals subjected to VFCA and eCPR associates with relevant neurofunctional outcome markers in two vulnerable brain regions, the hippocampus and cortex, assessed 14 days post-CA. Interestingly, we found that not the very early scoring of the neurological deficits (day 1–day 3), but only those assessed at day 4 and at day 5 post-ROSC correlated significantly with a reduced function of neuronal tissue and are useful for predicting the compromised long-term neurological outcome (Figure 9).

### 4.1. Animal Model

In a previous study using an experimental model of VFCA and conventional CPR by chest compression, we showed significantly reduced activities of HO in both the Hc and mC. Therefore, we questioned whether extracorporeal CPR, which we used in this study, would affect both regions similarly.

ECPR is recommended by current guidelines for selected patients with witnessed CA, bystander CPR within 5 min, a shockable initial rhythm, and an age below 70 years, if they cannot be resuscitated with conventional CPR [11]. The rationale behind eCPR is that fewer than 10% of patients survive with a good neurological outcome after 20 min of conventional CPR [41]. eCPR appears to offer improved neurological outcomes compared to conventional resuscitation in this patient population [42]. Therefore, the usage of eCPR has increased in recent years in industrialized regions with CA centers and established eCPR systems [43].

However, many research questions remain unanswered, and the demand for animal eCPR models with long-term neurological survival is very high. However, there is still a significant lack of small-animal models with long-term survival in particular [44]. Therefore, we developed an experimental VFCA model that more accurately mimicked clinical scenarios [44,45] using eCPR [31]. It is not clear to what extent the method of resuscitation employed during CA affects the degree of delayed neurological dysfunction resulting from ischemia–reperfusion injury [46]. We have previously shown that the use of eCPR following prolonged VFCA resulted in a 100% resuscitation rate and an 88% survival rate at 14 days compared to a 63% resuscitation rate and a 13% survival rate in a conventional CPR group [13]. Therefore, one aim of this study was to evaluate and to compare neurological outcomes in our newly established model using the previously identified neurofunctional outcome parameters. Additionally, we questioned whether the activity of an important mitochondrial enzyme complex, namely, OGDHC, which we have previously found compromised in a model of acute traumatic brain injury [28], would also be affected in our model. Finally, we aimed to understand which is the earliest time point for scoring neurological deficits with the potential to predict the delayed neuronal dysfunction that manifests at 2 weeks post-CA.

### 4.2. Increased Neuroinflammation and Gliosis in the Hippocampus Two Weeks After VFCA and eCPR

In line with the data from our previous study, in which we applied conventional CPR via chest compression [9], we found a significantly increased mRNA expression of HO-1 and TNF-R1 in Hc tissue, which indicates increased cell stress and ongoing neuroinflammation in this brain region. Given the particular relevance of HO-2 for maintaining neuronal survival and oxidative stress response [47], we additionally analyzed the distribution of the HO-2 signal in brain slices by immunohistochemistry. In the Hc, we found significantly decreased neuron-specific HO-2 signals, probably mirroring the neuronal loss that we reported elsewhere [9]. Simultaneously, we found an increased glia-specific HO-2 signal in hippocampus tissue. This increase was accompanied by increased Iba1 and GFAP staining, these being typical markers indicating glial activation, which was in accordance with our previous study [31]. It is likely that the increase in the glia-specific HO-2 signal, which we found in both our studies, reflects the increased activation and/or proliferation of glial cells in response to ischemia-induced neuronal loss, which has been shown in gerbil hippocampi 4 days after global ischemia [48]. Thus, the increased HO-2 signal provided by glial cells reflects the attempt to enhance tissue repair and cope with the increased oxidative stress. While in Hc all markers showed a clear association with the duration of CA, clear signs of neuroinflammation or neurodegeneration were not detected in the cortex, similarly to our previous study [9].

### 4.3. Neuronal Outcome Biomarkers Indicate Dysfunction and Degeneration of Neurons in the Hippocampus and Cortex Two Weeks After VFCA and eCPR

Despite the increased expression of HO-1 mRNA in hippocampus tissue and the strong increase in the glial HO-2 signal, hippocampal tissue displayed significantly lower HO activities. In neuronal tissues, HO-2 contributes nearly exclusively to the overall HO activity [49], and deletion of HO-2 is associated with a high susceptibility to different types of stress causing neuronal injury [50,51]. HO-2 KO mice show greater damage in neuronal tissue and extensive loss of neurons after traumatic brain injury [52]. Despite a strong HO-1 induction in this model, HO activity in HO-2 KO mice remained significantly lower than that of HO-2 wild type [52]. This indicates that HO-1 induction cannot compensate for a decrease in overall HO activity, which may be caused by loss of neurons, the cell type contributing predominantly to HO activity in neuronal tissue [53]. Different to our previous study, cortex tissue did not display decreased HO activities, although there was a trend of a decrease seen in the 8 min-CA animals compared to the 6 min-CA group. The slightly better situation revealed in the cortex might indicate that eCPR provokes less neurological damage as compared to conventional CPR. However, in the Hc, our findings correspond to those previously described [9], indicating a good reproducibility for both VFCA models. Thus, the global ischemic episode provoked by CA sets in motion subsequent pathologic sequelae, which appear to be relatively independent of the resuscitation method applied—CPR by chest compression or eCPR. The significantly lower heme-degrading capacity of Hc tissue, which we found in both models, indicates an increased susceptibility to oxidative stress and a compromised production of HO products that are essential for neurotransmission, modulation of synaptic activity, and neuroprotection [17], particularly under conditions of elevated glutamate levels [22]. We therefore suspected that another sensitive enzyme system, OGDHC, which plays an important role in controlling extracellular glutamate levels [28], is affected as well.

We are the first to show that OGDHC, the rate-limiting enzyme complex of the mitochondrial tricarboxylic acid cycle, displayed a significantly lower activity in the Hc two weeks after CA and resuscitation, which was additionally dependent on the CA duration. Albeit not significantly, OGDHC activity was also lower in mC tissue in the 8 min-CA group. Of note, in both tissues, Hc and mC, the activities of HO and OGDHC correlated significantly with each other within one brain region as well as between different regions of the brain within the animals. This shows that delayed CA-mediated derangements affect both enzyme systems simultaneously and both brain regions in a coordinated fashion. The positive correlation of the OGDHC-specific signal intensity with the HO enzyme activity within mC tissue supports this assumption further.

High levels of NO and subsequent nitrosative stress result from activated glial cells, which express iNOS in response to activation with different pro-inflammatory stimuli [28]. Elevated levels of NO in conjunction with oxidative stress are believed to cause neuronal death following ischemia and trauma and in other neurodegenerative diseases [54]. One group reported oxidative/nitrosative modifications of HO-1 and BVR protein occurring in the hippocampi of AD patients [55,56] and hypothesized that these modifications compromised the enzymatic function of both enzymes, which contributed to neurodegeneration. The underlying mechanism involves upregulation of iNOS and increased formation of NO and RONS [57] in activated astrocytes [58]. Indeed, in our previous study investigating the mechanisms of brain injury following aneurysmal subarachnoid hemorrhage, we showed that elevated levels of NO impair the activity of OGDHC [28]. However, in our model, at two weeks after CA, we found that levels of iNOS mRNA were unchanged in both brain regions, despite reactive astroglia in the Hc. Therefore, we do not have evidence to assume increased nitrosative stress operating in our model at this time point. Nevertheless, we cannot exclude that under other experimental/clinical settings excessive generation of NO may affect the activity of these enzymes. An excessive production of NO and formation of peroxynitrite, causing protein nitrosylation and potential protein dysfunction, was shown at 3 h after reperfusion in a model of focal brain ischemia [59]. In a rat model, using artery occlusion, iNOS expression was shown to rise in ischemic neuronal tissues during the first hours and to decline back to baseline levels during the subsequent days [58]. In human studies, elevated levels of NO were observed within one week after insult and normalized during the second week [60]. Since we did not find evidence of increased oxidative/nitrosative stress at the time point analyzed in our study, our findings suggest that the decrease in catalytic activities, which we found for HO and OGDHC, result less from damage to the enzyme systems and more from loss of neurons. In line with this hypothesis is the interesting finding that NOx levels were even lower in Hcs from animals subjected to CA compared to the sham animals. Since NO is an important neurotransmitter and produced by neurons [61], lower levels of NOx in the Hcs of 8 min-CA animals support the assumption of lower amounts of functional neurons within this brain region.

This interpretation also explains the decreased HO-2- and OGDHC-specific neuronal signals in brain regions from animals subjected to CA. As functional neurons are the cell types that display the highest HO-2 levels, it appears that the decrease in the activity of both enzyme systems in the Hcs of CA rats reflects the degree of neuronal loss or dysfunction in the brain. As chronic or long-term inhibition of OGDHC may cause calcium dysregulation, similar to that seen in Alzheimer’s disease [62], such processes may also underlie the neurodegeneration observed in our model. Interestingly, although the cortex appeared morphologically normal, neuronal OGDH levels were diminished in both regions, Hc and mC. Contrary to the determination of enzyme activity using homogenates, analyses using IHC allow focus on the cell type of interest. HO-2 is mainly expressed and OGDHC shows its highest activity levels in the neuron-rich cerebral cortex [63]. It is possible that this is the reason why IHC was more sensitive in determining neuron-specific loss of both enzyme systems in the mC in response to CA. The strong correlation of both enzyme activities with each other and with different brain regions shows that the degree of enzyme dysfunction, which we determined in the Hc, allows extrapolation on the strength of ROSC-mediated neuronal dysfunction that has affected the entire brain. The changes which we determined at 2 weeks after ROSC at the enzyme level most probably reflect an increased number of dysfunctional neurons or a loss of neurons in possibly all brain regions.

Given the multitude of factors driving post-CA brain injury, the determination of early measures for reliably predicting the long-term neuronal outcome in individuals resuscitated from CA remains a very difficult task [46]. We therefore scored the neurological deficits daily and questioned whether this was a useful parameter for predicting the delayed neuronal dysfunction found in the rat brains two weeks after ROSC, and, if it was, which time point was most suitable.

### 4.4. Scoring Neurological Deficits at 4 to 5 Days After CA Predicts Long-Term Neurological Outcome Following VFCA and Resuscitation by eCPR

Our model was associated with increased mortality during the initial 3 to 4 days in relation to the duration of CA. A study using a porcine CA model described several factors contributing to early death, such as massive neurological injury, hemodynamic instability, and multi-organ failure, resulting from systemic ischemia–reperfusion [64], conditions assumed to strongly affect NDS. So as not to confound the early scores of neurological deficits, we excluded all those rats that did not survive the entire 2-week period from the analyses.

Interestingly, only the earliest time point measured (24 h after CA) correlated significantly with the expression of HO-1 in the cortices from the 2-week survivors. This indicates that cortex tissue “remembers” for a long time the intensity of stress experienced during CA and resuscitation, which directly translates into the occurrence of neurological deficits appearing early after ROSC. In autopsy samples of human brains from patients with traumatic brain injury or focal ischemia which were analyzed at different time points, accumulation of HO-1-positive microglia/macrophages was determined as early as 6 h until 6 months after the ischemic insult [36]. However, neither HO-1 mRNA levels nor NDS at 24 h correlated with the degree of enzymatic dysfunction in neuronal tissue, and therefore neither are predictive for the neuronal loss in the Hc or mC seen two weeks after ROSC.

We assume that the primary damage to the brain tissue, which already showed at 6 h of reperfusion [65], is reversible to a certain degree because neurological deficits gradually declined throughout the initial days. In the 6 min-CA group, we found a continuous decline in neurological deficits during the initial 6 days, while in the 8 min-CA group the NDS remained unchanged from day 4 on. This indicates that recovery from acute stress needs about 3 to 4 days in our model. A rat model in which artery occlusion was used to induce cortical brain ischemia showed that the duration of motoric and cognitive impairment was dependent on the severity of the ischemic period. While permanent artery occlusion resulted in long-term neurological deficits, transient artery occlusion resulted in sensorimotor deficits, which gradually declined in the subsequent weeks. The severity of deficits also correlated with the induced infarct size evaluated after 30 days [66]. We found that the decrease in enzyme activity determined at 2 weeks after CA and resuscitation, which represented neuronal function in the Hc and mC, correlated significantly with the NDS determined 4 and 5 days after ROSC. In our opinion, this time point reflects the transition from the acute recovery phase to the phase characterized by lasting neuronal dysfunction. A mechanistic view of these steps is presented in Figure 9.

### 4.5. Clinical Implications

Out-of-hospital cardiac arrest still has a mortality rate of more than 90% [67]. The majority of patients who initially achieve ROSC later die due to neurological damage. For these patients, the only treatment option is therapeutic hypothermia, but no precise pharmaceuticals are currently available. Our study shows a strong inverse correlation of HO and OGDH activity (both potential treatment targets) with functional outcomes. This highlights the important role of oxidative stress and inflammation in the phase following ROSC and the need for appropriate treatment options.

Additionally, our data support that eCPR “patients” need more time to recover than conventional CPR patients. Even after 6 or 8 min of CA followed by immediate eCPR, day 3 (as recommended in the guidelines) was too early to detect an association with long-term neurological damage. Transferred to human patients, this would mean that much longer resuscitation times must be accompanied by much longer windows for neuroprognostication. In fact, Bartos et al. observed exactly this phenomenon in human eCPR patients [12].

### 4.6. Limitations of the Study

While this study provides important insights into the molecular and functional consequences of CA-induced brain injury, certain limitations must be acknowledged. First, although neurological deficits were assessed at multiple time points, enzyme dysfunction in the hippocampus and cortex was only evaluated at later stages, and its presence at earlier phases and its underlying mechanisms remain undetermined. Future investigations are planned to conduct longitudinal biochemical assessments at multiple time points and to address the potential role of mitochondrial dysfunction to clarify the underlying mechanisms.

Further, it is important to note that the use of young, healthy rats as experimental models limits the direct extrapolation of these findings to the heterogeneous population of human CA patients, which often includes older individuals and patients with pre-existing comorbidities. Therefore, research that takes into account the clinical complexity of human CA patients is needed to validate and extend these observations.

### 4.7. Conclusions

Currently, it is unknown how large or wide the therapeutic windows are for minimizing primary and secondary brain injury post-CA. However, our data, obtained using an animal model to better mimic clinical scenarios, suggest that not immediately after, but day 4 to 5 post-CA is the critical window for identifying individuals at risk of long-lasting adverse neurological outcomes. These data might also indicate that these initial days (up to day 5) are most suitable for potential therapy, possibly accompanied by supplementation with HO products or thiamin, for targeting HO and OGDHC dysfunction and moderating secondary brain injury in individuals who survive CA.

## Figures and Tables

**Figure 1 biomolecules-15-00732-f001:**
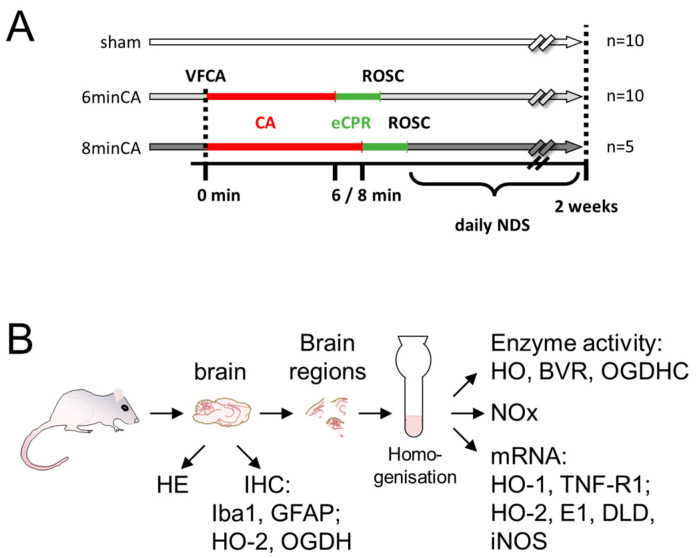
Experimental setup. (**A**) Rats underwent instrumentation only (sham, open bars) or 6 (light-grey bars) or 8 min (dark-grey bars) of cardiac arrest (6 min-CA, 8 min-CA; red bars), followed by resuscitation (eCPR, green bars) until return of spontaneous circulation (ROSC). Neurologic deficit scores were recorded daily (NDS). Rats surviving 2 weeks were further analyzed (sham, n = 10; 6 min-CA, n = 10; and 8 min-CA, n = 5). (**B**) One brain half was used for histological examination (hematoxylin–eosin staining (HE)) and immunohistochemistry (IHC) for heme oxygenase (HO)-2, ionized calcium-binding adapter molecule 1 (Iba-1), and glial fibrillary acidic protein (GFAP), as well as 2-oxoglutarate dehydrogenase (OGDH, subunit E1). Regions cut from the other half were homogenized and analyzed for enzyme activity (HO, BVR, and OGDHC), levels of NOx, and gene expression for HO-1, tumor necrosis factor receptor 1 (TNF-R1), HO-2, subunit E1, and DLD (dihydrolipoyl dehydrogenase) of OGDHC, as well as inducible NO synthase (iNOS).

**Figure 2 biomolecules-15-00732-f002:**
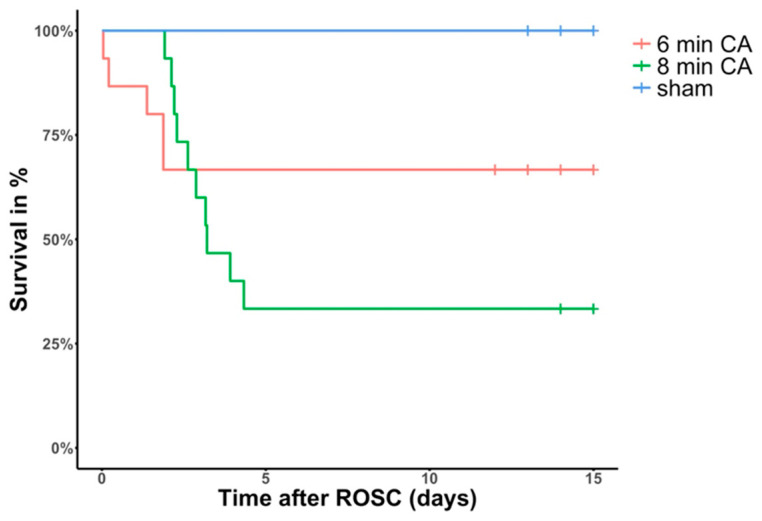
Kaplan–Meier analysis of cumulative survival two weeks after ROSC (return of spontaneous circulation). Red line = 6 min cardiac arrest (CA) animals, green line = 8 min-CA animals. The 10 sham-operated animals (blue line) all survived until the study endpoint. Tick marks in the respective curves indicate animals, which did not die, but were sacrificed earlier or later than the intended 14 days due to logistical reasons.

**Figure 3 biomolecules-15-00732-f003:**
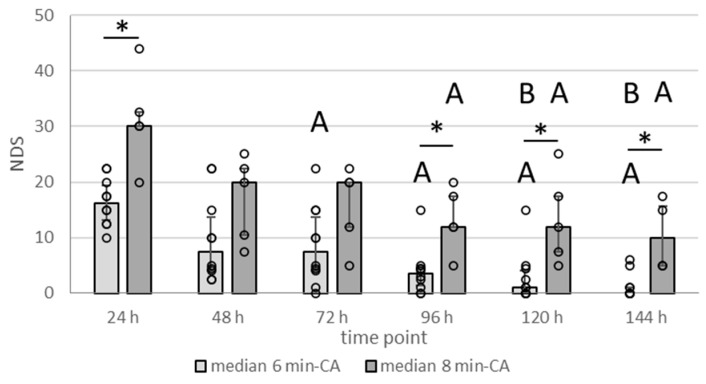
Scores of neurological deficits in function at various time points in animals subjected to 6 min or 8 min of CA followed by extracorporeal CPR. Neurologic deficit scores (NDSs) were recorded daily from day 1 (24 h) until day 6 (144 h), as described in Section 2.2, including only 2-week survivors. The higher the value, the bigger the neurological impairment. Sham animals that displayed an NDS of 0 (no impairment) throughout the observation period are not shown. Data are displayed as bars (6 min-CA group (n = 10), light grey; 8 min-CA (n = 5), dark grey) indicating medians and 1st (25%) and 3rd (75%) quartiles together with single values as open dots. Differences between 6 min-CA and 8 min-CA groups at a single time point are indicated (* *p* < 0.05, Wilcoxon test followed by Dunn–Bonferroni correction). Letters indicate where the comparison between time points was significant (*p* < 0.05 using the paired two-factor Friedmann test) as A = compared to 24 h, and B = compared to 48 h values for the respective groups.

**Figure 4 biomolecules-15-00732-f004:**
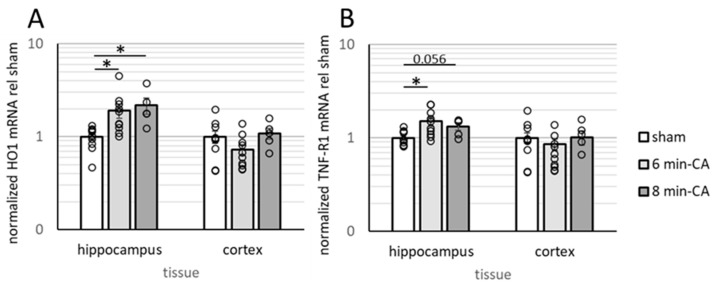
Markers for cell stress and neuroinflammation in Hcs and mCs of rats two weeks after 6 or 8 min CA. Brain sections from sham-operated rats (sham, white bars, n = 10) and rats subjected to CA for 6 min (light-grey bars, n = 10) and 8 min (dark-grey bars, n = 5) were processed as described in Section 2.4 and subjected to gene expression analysis as described in 2.8. For each region, the normalized expression levels of (**A**) HO-1 and (**B**) TNF-R1 in each animal were calculated relative to the sham means. Averaged group data are visualized on a log10 scale as means indicating SEMs and single values as open dots. Differences between groups were analyzed using the Kruskal–Wallis test followed by Dunn–Bonferroni correction. Significant differences between groups are shown (* *p* < 0.05) and *p*-values for near-significant differences are indicated.

**Figure 5 biomolecules-15-00732-f005:**
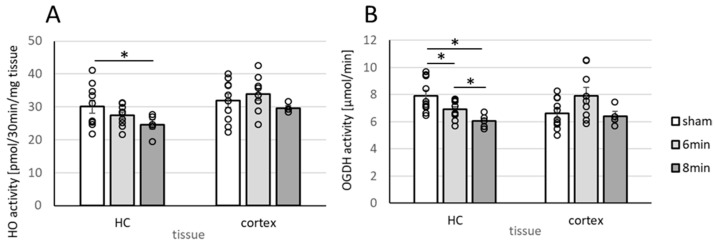
Activity of enzymes essential for neuronal function in Hcs and mCs of rats two weeks after 6 or 8 min CA. Samples were processed as described in Section 2.4 and the legend of Figure 4. Analysis of enzyme activity was performed as described in Section 2.5 and Section 2.6 and given as pmol bilirubin produced in 30 min/mg tissue for HO (**A**) or as µmol NADH formed per min/mg tissue for OGDHC (**B**). Description of the groups is the same as in Figure 4. Averaged group data are displayed as means indicating SEMs and single values as open dots. Differences between groups were analyzed using the two-tailed heteroscedastic Student’s *t*-test. Significant differences between groups are shown (* *p* < 0.05).

**Figure 6 biomolecules-15-00732-f006:**
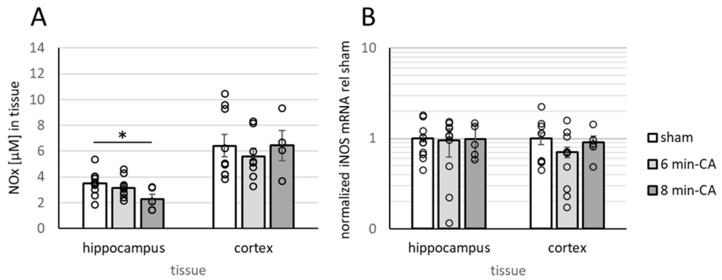
Quantification of NOx levels and expression of inducible NO synthase in Hcs and mCs of rats two weeks after 6 or 8 min CA. Samples were processed as described in Section 2.4. Analysis of NOx (**A**) was performed as described in Section 2.7, and determination of iNOS expression levels (**B**) was performed as described in Section 2.6 and corresponds to the information given in the legend to Figure 4, including the description of groups. Averaged group data are displayed as means indicating SEMs and single values as open dots. Differences between groups were analyzed using the two-tailed heteroscedastic Student’s *t*-test (NOx), or the Kruskal–Wallis test, followed by Dunn–Bonferroni correction (iNOS), indicating significant differences (* *p* < 0.05).

**Figure 7 biomolecules-15-00732-f007:**
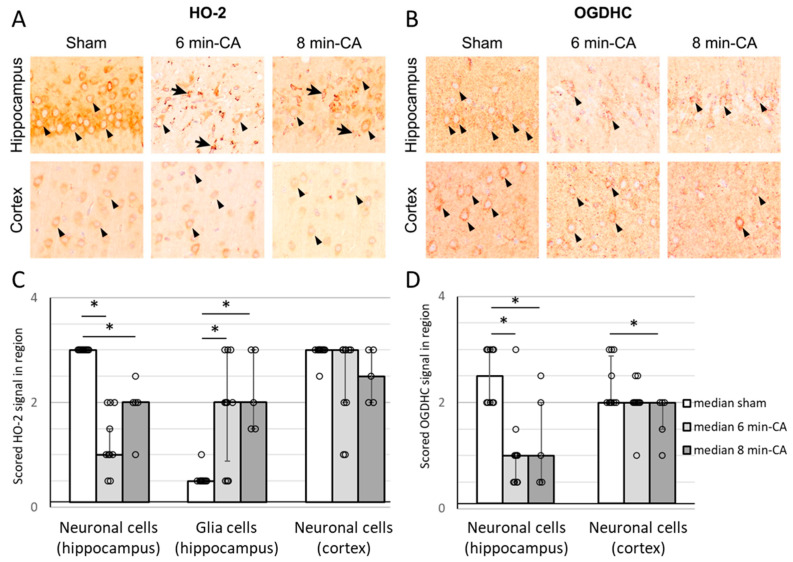
Semi-quantitative determination of the HO-2 and OGDH signals in neurons and glial cells of Hc and mC tissue of rats 14 days after CA. Brains were processed for immunohistochemical analysis performed as described in Section 2.3. Representative images of (**A**) HO-2- and (**B**) OGDH- specific signals. (**A**) top: Intense HO-2 expression in the pyramidal neurons (arrowheads) of the Hcs of sham animals, while neurons expressing HO-2 (arrowheads) are significantly reduced in CA animals. Increased HO-2 expression in glial cells (arrows) in the Hcs of CA-animals. (**A**), bottom: No significant differences in the HO-2 expression in neurons (arrowheads) of mCs. (**B**), top: Intense OGDH expression in the pyramidal neurons (arrowheads) of the Hcs of sham animals, while neurons expressing OGDH (arrowheads) are significantly reduced in CA animals. (**B**) bottom: Intense OGDH expression in neurons (arrowheads) of the mCs in sham and 6 min-CA animals, while neurons expressing OGDH (arrowheads) are significantly reduced in 8 min-CA animals compared to sham animals. Scoring results are displayed as bars for (**C**) HO-2 signals and (**D**) OGDH (sham group (n = 10), white; 6 min-CA group (n = 10), light grey; 8 min-CA (n = 5), dark grey) indicating medians and 1st (25%) and 3rd (75%) quartiles together with single values as open dots. Differences between groups are indicated (* *p* < 0.05; Kruskal–Wallis test, followed by Dunn–Bonferroni correction).

**Figure 8 biomolecules-15-00732-f008:**
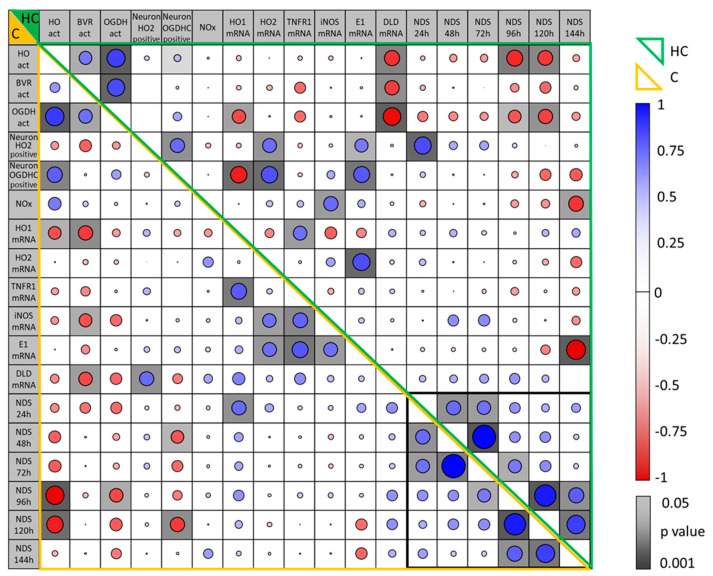
Correlation between parameters determined in the brain regions, hippocampus and motor cortex, in rats two weeks after CA. Enzyme activities of HO, BVR, and OGDHC; neuronal expression of HO and OGDHC; and mRNA expression levels measured at two weeks post-CA (6 min, n = 10; 8 min, n = 5) were correlated within each brain region (hippocampus, HC, green frame; cortex, C, yellow frame) and with neurologic deficits scores (NDSs) at early time points (24 h–144 h, black frame) post-CA. Correlation coefficients (Spearman’s Rho) are visualized by the size of the diameter of the circle in each field (blue, R > 0; red, R < 0). Levels of significance (*p* < 0.05) are indicated for correlations in the cells (shades of grey).

**Figure 9 biomolecules-15-00732-f009:**
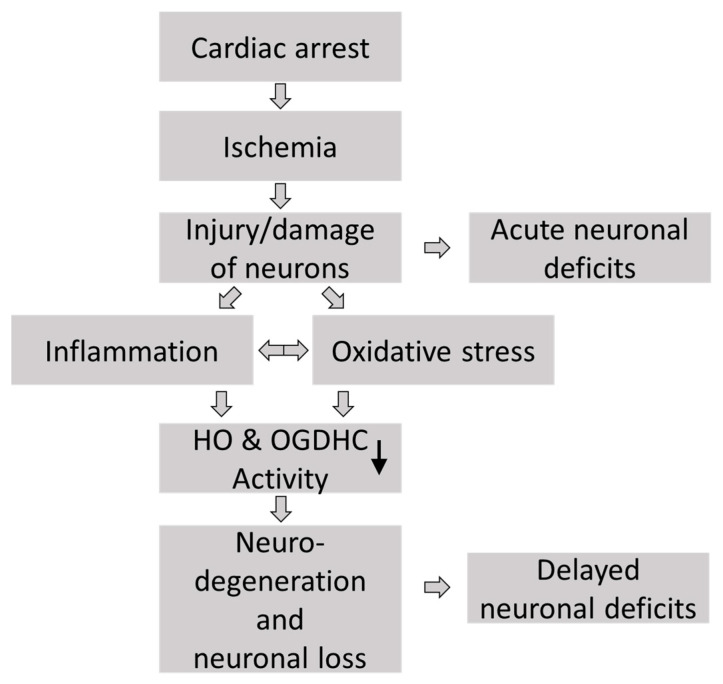
Graphical visualization of the hypothesized pathologic sequelae in brains of long-term CA survivors and their association with recordable neurological deficits. Cardiac arrest induces global ischemia, which results in ischemia–reperfusion injury following resuscitation. The immediate and acute injury to vulnerable neurons is associated with acute (and partially reversible) neuronal deficits, which is known as “primary” brain injury. As a response to primary damage, subsequent pathological events, characterized by enhanced inflammation and increased oxidative stress, are set in motion, which turn into long-lasting neurological deficits due to neurological loss and ongoing neurodegeneration.

**Table 1 biomolecules-15-00732-t001:** Neurological deficit score (NDS) in rats.

Parameter		Score ^1^		Characteristics		
		Maximal points (worst condition)	Percentage (max 100%)	Description		
General deficit	Consciousness	20	20	20 (comatose)	10 (somnolent)	0 (normal)
	Respiration	20	20	20 (abnormal breathing)		0 (normal breathing)
Cranial nerve reflexes deficit	Olfactory (sniffing food)	4	4	4 (no reaction)		0 (normal)
	Vision (follows hand)	4	4	4 (no reaction)		0 (normal)
	Corneal reflex	4	4	4 (absent)		0 (present)
	Whisker Movement	4	4	4 (no whisker movement)		0 (normal)
	Hearing (turns to clapped hands)	4	4	4 (no reaction)		0 (normal)
Motor deficit	Motor	10	10	2.5 points for motoric problems for each affected extremity		0 (normal)
Sensory deficit	Sensory	10	10	2.5 points for sensory loss for each affected extremity		0 (normal)
Coordination deficit	Travel ledge	5	5	5 (no ability to walk on a beam)	2.5 (ability to walk on a beam with some help or for shorter periods)	0 (physiological behaviour)
	Placing test (front paws reaching when lifted from ground by tail)	5	5	5 (absent)		0 (present)
	Righting reflex (attempting to right self when placed on back)	5	5	5 (absent)		0 (present)
	Stop at table edge	5	5	5 (absent)	2.5 (animal recognises table edge, but is too weak to prevent falling down)	0 (present)

^1^ A normal rat has an NDS of 0 points (0%); a (brain) dead rat has an NDS of 100 points [32].

**Table 2 biomolecules-15-00732-t002:** Correlation of HO and OGDHC activity of Hc and mC tissue in rats two weeks after CA.

Parameter 1	Parameter 2	Correlation Coefficient
HO activity in Hc	HO activity in mC	0.895 **
HO activity in Hc	OGDHC activity in Hc	0.744 **
HO activity in Hc	OGDHC activity in mC	0.819 **
HO activity in mC	OGDHC activity in Hc	0.699 **
HO activity in mC	OGDHC activity in mC	0.762 **
OGDHC activity in Hc	OGDHC activity in mC	0.512

** *p* < 0.01 (Spearman’s Rho).

**Table 3 biomolecules-15-00732-t003:** NDS at 1, 4, and 5 days after resuscitation correlated with compromised functional outcomes in the cortex and hippocampus after 14 days.

Functional Outcome After 2 Weeks	NDS at 24 h	NDS at 96 h	NDS at 120 h
Cortex HO-1 mRNA	0.59 *	0.45	0.31
Cortex HO activity [pmolBR/30 min/mg tissue]	−0.36	−0.75 **	−0.68 **
Cortex OGDHC activity [µmol/min/mg tissue]	−0.46	−0.56 *	−0.51
Hippocampal HO activity [pmol/30 min/mg tissue]	−0.2	−0.65 **	−0.61 *
Hippocampal OGDHC activity [µmol/min/mg tissue]	−0.42	−0.52 *	−0.59 *

* *p* < 0.05; ** *p* < 0.01 (Spearman’s Rho).

## Data Availability

The original contributions presented in this study are included in the article/Appendix A. Further inquiries can be directed to the corresponding author(s).

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
