# Peer review of "Neurologic Deficit Score at 4–5 Days Post-eCPR Predicts Long-Term Brain Dysfunction in Rats Following Cardiac Arrest"

_biomolecules, 2025, doi:10.3390/biom15050732_

Round 1
Reviewer 1 Report
Comments and Suggestions for Authors
In this manuscript, Weihs et al. reported on the neuroprotective effects of both heme oxygenase (HO) and oxoglutarate dehydrogenase complex in rats exposed to cardiac arrest. The study has been well conducted, methods are appropriate and updated, and the results support the conclusion drawn. This reviewer has few suggestions, whose only purpose is to further improve the quality of the manuscript.
Abstract. A sentence that recapitulate the main results should be added at the end of the text.
Introduction. In my opinion, this section is too long and should be resized. As far as the effects of HO isozymes in the brain, I suggest the Authors to mention the paracrine effects of carbon monoxide that contribute to the proinflammatory effects of this enzyme (e.g., PMID 9149104).
Results. I suggest leaving in Figure 5 only the asterisks related to statistical significance. The values of p>0.05 should be erased. Previous studies have shown that, under pro-oxidant conditions, biliverdin reductase undergoes oxidative/nitrosative modifications affecting its activity. For this reason, I would smooth the sentence in lines 413-414 and discuss this potential issue in the Discussion Section.
Discussion. The sentence in lines 559-560 is an overstatement. HO-2 is mainly involved in the physiological turnover of heme and intracellular gas sensing, and not in the cell stress response. The sentence in lines 578-579 must be amended; indeed, hippocampus is a brain area endowed with significant HO-1 stain (PMID 9131263). In order to corroborate their findings, the Authors may want to consider the possibility of oxidative/nitrosative post-translational modificantions occurring in HO-1 under pro-oxidant conditions and affecting its activity .
Throughout the text. "Motor cortex" instead of "motoric cortex".
Author Response
We thank the Reviewer 1 for the critical evaluation and for pointing towards the strength of our study! We are happy to consider the comments and are currently trying to improve the manuscript. Please find our reply to the specified comments below:
In this manuscript, Weihs et al. reported on the neuroprotective effects of both heme oxygenase (HO) and oxoglutarate dehydrogenase complex in rats exposed to cardiac arrest. The study has been well conducted, methods are appropriate and updated, and the results support the conclusion drawn. This reviewer has few suggestions, whose only purpose is to further improve the quality of the manuscript.
Comment 1: Abstract. A sentence that recapitulate the main results should be added at the end of the text.
Answer to comment 1: We have complemented the abstract as follows: ‘Not the immediate, but NDS at day 4–5, strongly correlated with the delayed CA-mediated enzymatic dysfunction determined in hippocampus. This finding highlights this time-point for identifying at-risk individuals and suggests a prolonged therapeutic intervention lasting at least until 4 days post-CA’
Comment 2: Introduction. In my opinion, this section is too long and should be resized.
Answer to comment 2: We shortened the introduction. Please refer to the uploaded revised version of the manuscript.
Comment 3: As far as the effects of HO isozymes in the brain, I suggest the Authors to mention the paracrine effects of carbon monoxide that contribute to the proinflammatory effects of this enzyme (e.g., PMID 9149104).
Answer to comment 3: We thank reviewer 1 for the valuable suggestion of pointing towards the paracrine effects of carbon monoxide. We have added the following sentence to the introduction section: ‘However, prolonged HO-1 activation may potentially cause harm via the increased generation of CO, which may contribute to ischemia-triggered neuroinflammation in-volving activation of cyclooxygenase-2(1).’
Comment 4: Results. I suggest leaving in Figure 5 only the asterisks related to statistical significance. The values of p>0.05 should be erased.
Answer to comment 4: done
Comment 5: Previous studies have shown that, under pro-oxidant conditions, biliverdin reductase undergoes oxidative/nitrosative modifications affecting its activity. For this reason, I would smooth the sentence in lines 413-414 and discuss this potential issue in the Discussion Section.
Answer to comment 5: We thank reviewer 1 for this point. For clarity reasons, we inform that the text in line 413-414 refers to our own findings. We analyzed both, the mRNA expression of BVRA and the activity of BVR (by investigating the capacity of the tissue homogenate to convert biliverdin into bilirubin). We did not find any differences between groups (see figure 1 below).
|
Fig 1: Activity of Biliverdin reductase (BVR) in Hc and mC of rats two weeks after 6 or 8 min-CA. Samples were processed as described in the manuscript in section 2.4 and the legend of Fig. 4. Analysis of enzyme activity was done as described in 2.5. Instead of using hemin, homogenates were supplemented with biliverdin (200 µmol) and the produced bilirubin was quantified as described (2.5). Enzyme activity is given as pmol bilirubin produced in 30 min/mg tissue. Averaged group data are displayed as means indicating SEM and single values as open dots. |
Therefore, our data strongly suggest that BVR protein was not subjected to any changes affecting its activity. In conjunction with the unchanged levels of inducible NO synthase (iNOS) mRNA expression and the levels of NO derivatives (NOx), which were also not elevated (and even significantly lower in hippocampus tissue of rats subjected to CA) in our study, we do not have evidences to assume an increased nitrosative stress in our model at this time point. Nevertheless, to make this point clearer, we have changed the discussion section and by adding the following text:
‘One group reported oxidative/nitrosative modifications of HO-1 and BVR protein occurring in hippocampus of AD patients(2, 3), and hypothesized that these modifications compromised the enzymatic function of both enzymes, which contributed to neurodegeneration. The underlying mechanism involves upregulation of iNOS and increased formation of NO and RONS(4) in activated astrocytes(14). Indeed, in our previous study investigating the mechanisms of brain injury following aneurysmal subarachnoid hemorrhage we showed that elevated levels of NO impair activity of OGDHC. However, in our model at two weeks after CA, we found levels of iNOS mRNA unchanged in both brain regions, despite reactive astroglia in Hc. Therefore, we do not have evidences to assume an increased nitrosative stress operating in our model at this time point. Nevertheless, we cannot exclude excessive generation of NO in response to CA under other experimental/clinical settings that affect the activity of these enzymes.’
And further: ‘Since we did not find evidence of increased oxidative/nitrosative stress at the time point analyzed in our study, our findings suggest that the decrease in catalytic activities, which we found for HO and OGDHC. At this late time point, it appears to result less from damage to the enzyme systems, but more from loss of neurons. In line with this hypothesis is the interesting finding of in our study that NOx levels were even lower in Hc from animals subjected to CA compared to sham animals.’ The latter is additional evidence that nitrosative modification of enzymes is not a predominant mechanism occurring in our model.
In order to avoid any confusion, we adapted the abstract as follows: ‘Two weeks post-CA, neuroinflammatory and neurodegenerative markers (HO-1, TNF-R1, Iba1, GFAP) were elevated in the hippocampus, while HO and 2-oxoglutarate dehydrogenase complex activities were reduced in all rats, indicating a decreased anti-oxidative and a decreased mitochondrial capacity for metabolizing glutamate.’
Comment 6: Discussion. The sentence in lines 559-560 is an overstatement. HO-2 is mainly involved in the physiological turnover of heme and intracellular gas sensing, and not in the cell stress response. The sentence in lines 578-579 must be amended; indeed, hippocampus is a brain area endowed with significant HO-1 stain (PMID 9131263).
Answer to comment 6: We respectfully disagree with some of the statements made by reviewer 1. While it is accurate that the hippocampus exhibits some HO-1 expression, particularly in regions like the dentate gyrus, HO-1 levels are generally low under physiological conditions. This point is also clearly stated by the review cited by the reviewer(6), see page 525 and table 1). HO-1 has been shown to increase its expression significantly in response to stressors, particularly oxidative stress(7) and substances inducing excitotoxicity(8) The induced HO-1 exerts cytoprotection(9) mainly via an increased heme removal and the generation of HO products in concert with biliverdin reductase (10) These products are also generated by HO-2. It can be expected that also HO-2 inherently consists of cytoprotective and antioxidant properties. This was indeed demonstrated in a mouse model lacking HO-2. These mice, when subjected to focal cerebral ischemia, induced via middle cerebral artery occlusion, exhibited larger infarct volumes and increased neuronal apoptosis compared to wild-type controls(11) The same authors also showed the antioxidative property of HO-2 using in vitro studies. Neuronal cultures obtained from HO-2 knockout mice, showed elevated reactive oxygen species (ROS) production and increased cell death when exposed to oxidative agents like NMDA or glutamate, compared to cultures from wild-type mice. Given the fact that HO-2 is the major contributor to the overall HO activity in neuronal tissue (as has been described in the review of Maines, 1997:(6) we assume that a decreased enzymatic activity not only results in lower production of neurotransmitter but also reduces the antioxidative defense. We really hope that we have convinced the reviewer with the above arguments.
Comment 7: In order to corroborate their findings, the Authors may want to consider the possibility of oxidative/nitrosative post-translational modificantions occurring in HO-1 under pro-oxidant conditions and affecting its activity.
Answer to comment 7: Reviewer 1 pointed towards the relevance of increased oxidative/nitrosative stress that represents one important mechanisms underlying the pathogenesis of neurodegenerative disorders, such as Alzheimer disease. We thank reviewer 1 for the relevant suggestion. Since we addressed this point already in our response to comment 5, we politely ask the reviewer 1 to refer to this part and again we hope that the reviewer accepts our arguments.
Comment 8: Throughout the text. "Motor cortex" instead of "motoric cortex".
Answer to comment 8: done
References
- Mancuso C, Pistritto G, Tringali G, Grossman AB, Preziosi P, Navarra P. Evidence that carbon monoxide stimulates prostaglandin endoperoxide synthase activity in rat hypothalamic explants and in primary cultures of rat hypothalamic astrocytes. Brain Res Mol Brain Res. 1997;45(2):294-300.
- Barone E, Di Domenico F, Sultana R, Coccia R, Mancuso C, Perluigi M, et al. Heme oxygenase-1 posttranslational modifications in the brain of subjects with Alzheimer disease and mild cognitive impairment. Free Radic Biol Med. 2012;52(11-12):2292-301.
- Barone E, Di Domenico F, Cenini G, Sultana R, Coccia R, Preziosi P, et al. Oxidative and nitrosative modifications of biliverdin reductase-A in the brain of subjects with Alzheimer's disease and amnestic mild cognitive impairment. J Alzheimers Dis. 2011;25(4):623-33.
- Barone E, Di Domenico F, Cenini G, Sultana R, Cini C, Preziosi P, et al. Biliverdin reductase--a protein levels and activity in the brains of subjects with Alzheimer disease and mild cognitive impairment. Biochim Biophys Acta. 2011;1812(4):480-7.
- Weidinger A, Milivojev N, Hosmann A, Duvigneau JC, Szabo C, Törö G, et al. Oxoglutarate dehydrogenase complex controls glutamate-mediated neuronal death. Redox Biol. 2023;62:102669.
- Maines MD. The heme oxygenase system: a regulator of second messenger gases. Annu Rev Pharmacol Toxicol. 1997;37:517-54.
- Dwyer BE, Nishimura RN, Lu SY. Differential expression of heme oxygenase-1 in cultured cortical neurons and astrocytes determined by the aid of a new heme oxygenase antibody. Response to oxidative stress. Brain Res Mol Brain Res. 1995;30(1):37-47.
- Ahmad AS, Zhuang H, Doré S. Heme oxygenase-1 protects brain from acute excitotoxicity. Neuroscience. 2006;141(4):1703-8.
- Arai-Gaun S, Katai N, Kikuchi T, Kurokawa T, Ohta K, Yoshimura N. Heme oxygenase-1 induced in muller cells plays a protective role in retinal ischemia-reperfusion injury in rats. Invest Ophthalmol Vis Sci. 2004;45(11):4226-32.
- Baranano DE, Rao M, Ferris CD, Snyder SH. Biliverdin reductase: a major physiologic cytoprotectant. Proc Natl Acad Sci U S A. 2002;99(25):16093-8.
- Doré S, Sampei K, Goto S, Alkayed NJ, Guastella D, Blackshaw S, et al. Heme oxygenase-2 is neuroprotective in cerebral ischemia. Mol Med. 1999;5(10):656-63.
- Uittenbogaard M, Chiaramello A. Mitochondrial biogenesis: a therapeutic target for neurodevelopmental disorders and neurodegenerative diseases. Curr Pharm Des. 2014;20(35):5574-93.
- Scarpulla RC. Transcriptional activators and coactivators in the nuclear control of mitochondrial function in mammalian cells. Gene. 2002;286(1):81-9
- Iadecola C, Zhang F, Xu S, Casey R, Ross ME. Inducible nitric oxide synthase gene expression in brain following cerebral ischemia. J Cereb Blood Flow Metab. 1995;15(3):378-84.
Reviewer 2 Report
Comments and Suggestions for Authors
Weihs, W & Stommel, AM., et al tried to elucidate how often Cardiac arrest (CA) survivors experience long-term neurological deficits, and how the extent of its impact on the brain. The authors showed that in a rat model, survival rates decreased with longer CA durations (67% for 6 minutes vs. 33% for 8 minutes), and exhibited neurological impairments. However, the deficits improved within 3–4 days in the 6-minute group, the 8-minute group had worse outcomes until day 14. Moreover, elevated neuroinflammatory and neurodegenerative markers indicated oxidative stress and mitochondrial dysfunction.
The authors also showed the neurological deficit scores at days 4–5 strongly correlated with later enzymatic dysfunction in Hc and mC brain regions, suggesting a critical window for therapeutic intervention.
The authors pointed out 2 key questions in this study is that whether the severity of neurological deficits within the first week correlates with later reductions in key enzymatic activities (heme oxygenase [HO] and 2-oxoglutarate dehydrogenase complex [OGDHC]) in the hippocampus and cortex. Additionally, it seeks to identify the most predictive time point for neurological deficit scoring (NDS) to determine which deficits signal sustained neuronal dysfunction.
The objectives of this study carry significant importance; however, a pivotal lacuna exists concerning the escalating prevalence of aging as cardiac arrest is mostly experienced by the middle to older age group. The authors need to mention the potential limitation in extrapolating the findings to older age cohorts.
The experiments designed for this study are justified and the results are significant, but need a few more experiments to support the hypothesis. The introduction and the discussion were written clearly with proper information and references. High appreciation should go to the authors as they mention ethics in the methods.
Nonetheless, the article seemed to possess good value toward the relationship between the decline in enzyme activity two weeks after cardiac arrest correlated with neurological deficit scores at days 4–5, marking the transition from acute recovery to lasting neuronal dysfunction.
Overall, the clarity of the text needs readjustments. The manuscript has minor typographical and grammatical errors. The results and the figures were consistent. The quantitative analyses are much appreciated but need more careful revision based on the significance of the data. In general, the manuscript can accomplish the caliber of quality for consideration for publication in the Journal of “Biomolecules” with minor changes. The authors are advised to consider the comments below:
Comments:
- The title of the study is too lengthy. Please make it coherent and concise.
- Figure 1 – Please mention in the figure legend the specific brain regions used for the analysis of enzymatic activity.
- The authors need to mention the constraints, in the most crucial age group (the older age cohorts).
- In the portion – Analysis of gene expression – the term ‘described elsewhere’ is not acceptable. Please provide an appropriate reference.
- I would request the authors to provide the IHC staining images of Iba1 and GFAP in the supplementary.
- Please describe the procedure for motor and sensory deficits in the methods section.
- Please provide the specific area or location in the brain hippocampus of the IHC for determination of the HO-2- and OGDH-signals
- The authros need to provide a few IHC of oxidative stress and mitochondrial markers or provide data on mitochondrial oxygen comsumtion rate using either Oroboros or SeaHorse experiments.
Author Response
We thank the Reviewer 2 for the critical evaluation and for pointing towards the strength of our study! We are happy to consider the comments and are currently trying to improve the manuscript. Please find our reply to the specified comments below:
Weihs, W & Stommel, AM., et al tried to elucidate how often Cardiac arrest (CA) survivors experience long-term neurological deficits, and how the extent of its impact on the brain. The authors showed that in a rat model, survival rates decreased with longer CA durations (67% for 6 minutes vs. 33% for 8 minutes), and exhibited neurological impairments. However, the deficits improved within 3–4 days in the 6-minute group, the 8-minute group had worse outcomes until day 14. Moreover, elevated neuroinflammatory and neurodegenerative markers indicated oxidative stress and mitochondrial dysfunction.
The authors also showed the neurological deficit scores at days 4–5 strongly correlated with later enzymatic dysfunction in Hc and mC brain regions, suggesting a critical window for therapeutic intervention.
The authors pointed out 2 key questions in this study is that whether the severity of neurological deficits within the first week correlates with later reductions in key enzymatic activities (heme oxygenase [HO] and 2-oxoglutarate dehydrogenase complex [OGDHC]) in the hippocampus and cortex. Additionally, it seeks to identify the most predictive time point for neurological deficit scoring (NDS) to determine which deficits signal sustained neuronal dysfunction.
The objectives of this study carry significant importance; however, a pivotal lacuna exists concerning the escalating prevalence of aging as cardiac arrest is mostly experienced by the middle to older age group. The authors need to mention the potential limitation in extrapolating the findings to older age cohorts.
The experiments designed for this study are justified and the results are significant, but need a few more experiments to support the hypothesis. The introduction and the discussion were written clearly with proper information and references. High appreciation should go to the authors as they mention ethics in the methods.
Nonetheless, the article seemed to possess good value toward the relationship between the decline in enzyme activity two weeks after cardiac arrest correlated with neurological deficit scores at days 4–5, marking the transition from acute recovery to lasting neuronal dysfunction.
Overall, the clarity of the text needs readjustments. The manuscript has minor typographical and grammatical errors. The results and the figures were consistent. The quantitative analyses are much appreciated but need more careful revision based on the significance of the data. In general, the manuscript can accomplish the caliber of quality for consideration for publication in the Journal of “Biomolecules” with minor changes. The authors are advised to consider the comments below:
Comments:
Comment 1: The title of the study is too lengthy. Please make it coherent and concise.
Answer to comment 1: We have changed the title into: “Neurologic Deficit Score at 4–5 Days after Resuscitation Predicts Long-Term Brain dysfunction of Rats Following Cardiac Arrest"; Subtitle:
“Decreased Activities of Heme Oxygenase and Oxoglutarate Dehydrogenase Complex in Hippocampus Two Weeks Post-Insult". To our opinion this title better reflects the major findings of the study.
Comment 2: Figure 1 – Please mention in the figure legend the specific brain regions used for the analysis of enzymatic activity.
Answer to comment 2: We added the information requested by reviewer 2 and changed the manuscript accordingly. “In neurons of the hippocampal CA1 region and the motor cortex, protein expression of HO-2 and OGDH was rated semi-quantitatively in IHC-stained sections. Furthermore, HO-expression in glial cells was assessed semi-quantitatively in the hippocampal CA1 region.”
Comment 3: The authors need to mention the constraints, in the most crucial age group (the older age cohorts).
Answer to comment 3: We have mentioned this point by adding the following sentences into the discussion section: ‘Cardiac arrest is a major public health issue and affects particularly the elderly population, aged 65 years or older. However, favorable neurological outcomes among patients resuscitated from CA significantly decreases with age, which is generally attributed to increased comorbidity, and overall worse health of the elderly compared to younger people (PMID: 30924892).
Comment 4: The authors need to mention the potential limitation in extrapolating the findings to older age cohorts.
Answer to comment 4: We fully agree that the results obtained in this study cannot be extrapolated on the population of human patients, which are heterogeneous in terms of age and health state. We therefore acknowledge this limitation by adding a subchapter into the discussion section termed “Limitations of this study” and added the following sentences: ‘Further, it is important to note that the use of young, healthy rats as experimental model limits the direct extrapolation of these findings to the heterogeneous population of human CA-patients, which often includes older individuals and patients with pre-existing comorbidities. Therefore, research taking into account the clinical complexity of human CA-patients is needed to validate and extend these observations.’ Although this point was out of the scope of our study, we will definitely address it in our future projects.
Comment 5: In the portion – Analysis of gene expression – the term ‘described elsewhere’ is not acceptable. Please provide an appropriate reference.
Answer to comment 5: The original manuscript was supplemented with the respective reference. However, we reformulated the sentence to better position the reference.
Comment 6: I would request the authors to provide the IHC staining images of Iba1 and GFAP in the supplementary.
Answer to comment 6: We prepared the figure shown below (figure 2), which was incorporated into the supplementary material, and adjusted the respective sentence in the “Results” section of the manuscript: ‘Markers indicating activation of astrocytes/microglia cells (GFAP and Iba1) showed increased levels in the CA1 region of the Hc of CA animals (see supplementary Fig. S1), as was previously described in this model(29).’
|
Fig. 2. Representative images of the glial reaction in Hc and mC tissue of rats 14 days after CA. Immunohistochemistry was performed with antibodies against glial fibrillary acidic protein (GFAP) for detection of astrocytes (left panel) and ionized calcium binding adaptor molecule 1 (Iba1) for detection of microglia (right panel). The Hc shows increased signal intensity of both, astrocytes and microglia, after 6 and 8 min of CA compared to sham. The pyramidal layer of the hippocampal CA1 region is depicted by arrowheads (top). In contrast, the mC does not show any glial reaction after 6 and 8 min of CA compared to sham (bottom). Bars = 20µm. |
Comment 7: Please describe the procedure for motor and sensory deficits in the methods section.
Answer to comment 7: We added the following text to the Material and Methods section: “For the motor skills test, each limb was examined for its capacity of physiological movement. If no movement was possible, 2.5 points were given per limb, up to a maximum of 10. The remaining points for motor skills resulted from the points for travel ledge, placing test (front paws reaching when lifted from ground by tail) and righting reflex (attempting to upright itself when placed on the back). The sensory system of the extremities was tested by gentle stimulation with a needle and should be responded by retracting the limb or twitching the muscles.”
Comment 8: Please provide the specific area or location in the brain hippocampus of the IHC for determination of the HO-2- and OGDH-signals
Answer to comment 8: done
Comment 9: The authros need to provide a few IHC of oxidative stress and mitochondrial markers or provide data on mitochondrial oxygen comsumtion rate using either Oroboros or SeaHorse experiments.
Answer to comment 9: We appreciate the reviewer’s valuable suggestion regarding additional experiments. However, the present study was not designed to elucidate the mechanisms underlying the enzyme dysfunction, which we observed in hippocampus two weeks after CA. The scope of this study was to identify the most relevant time points for scoring neurological deficits predicting later enzyme dysfunction in the brain. Conducting the requested analyses would require a substantial expansion of the current project, including new animal ethics approvals and a new experimental series, particularly, because determination of mitochondrial function requires freshly prepared brain samples. Given the scope and timeline of the present study, we respectfully consider this beyond its framework. Additionally, we have no indications that dysfunctional mitochondria caused the decreased enzyme activity. In contrast, we think that loss of control over extracellular glutamate, as described previously(5)), rather than commonly assumed impaired energy metabolism, is the critical pathological manifestation of insufficient OGDHC activity, leading to neuronal death. Either way, conditions in which mitochondrial dysfunction plays a significant role in neurodevelopmental and neurodegenerative diseases, an increased mitochondrial biogenesis is considered a common compensatory mechanism (12) We therefore additionally determined mRNA expression of a marker for mitochondrial biogenesis (TFAM, an essential regulator of mtDNA copy number (13) in hippocampus tissue. Our data clearly show that TFAM is not increased indicating mitochondrial biogenesis was not stimulated (see figure 3. below), which is in line with the mRNA expression of the other nuclear coded mitochondrial proteins, OGDH (E2) and DLD (E3) and which supports our assumption that mitochondria do not appear dysfunctional in our samples at this time point.
To clarify this issue, we have adjusted the respective sentence in the abstract as follows: ‘Two weeks post-CA, neuroinflammatory and neurodegenerative markers (HO-1, TNF-R1, Iba1, GFAP) were elevated in the hippocampus, while HO and 2-oxoglutarate dehydrogenase complex activities were reduced in all rats, indicating a decreased anti-oxidative and a decreased mitochondrial capacity for metabolizing glutamate.’
Anyway, we fully agree on the importance of clarifying the mechanism underlying the decreased activity of OGDHC and HO, and we will address a possible impairment of mitochondrial function as a priority in future investigations. Additionally, we have supplemented the subchapter “Limitation of this study” with the following sentences: ‘While this study provides important insights into the molecular and functional consequences of CA-induced brain injury, certain limitations must be acknowledged. First, although neurological deficits were assessed at multiple time points, enzyme dysfunction in the hippocampus and cortex was only evaluated at later stages, and its presence at earlier phases and its underlying mechanisms remains undetermined. Future investigations are planned to conduct longitudinal biochemical assessments at multiple time points and to address the potential role of mitochondrial dysfunction to clarify the underlying mechanisms.’
|
Fig 3: Analysis of expression of mitochondrial proteins coding in the nucleus in hippocampus tissue of rats two weeks after 6 or 8 min-CA. Levels of mRNA, oxoglutarate dehydrogenase (OGDH), dihydro-lipoyl dehydrogenase, (DLD, syn. with E3) and a marker for mitochondrial biogenesis (regulator of mtDNA copy number, TFAM) were determined by qPCR in tissue homogenate, as described in the manuscript in section 2.9 and the legend of Fig. 4. Averaged group data are displayed as means indicating SEM and single values as open dots. |
- Mancuso C, Pistritto G, Tringali G, Grossman AB, Preziosi P, Navarra P. Evidence that carbon monoxide stimulates prostaglandin endoperoxide synthase activity in rat hypothalamic explants and in primary cultures of rat hypothalamic astrocytes. Brain Res Mol Brain Res. 1997;45(2):294-300.
- Barone E, Di Domenico F, Sultana R, Coccia R, Mancuso C, Perluigi M, et al. Heme oxygenase-1 posttranslational modifications in the brain of subjects with Alzheimer disease and mild cognitive impairment. Free Radic Biol Med. 2012;52(11-12):2292-301.
- Barone E, Di Domenico F, Cenini G, Sultana R, Coccia R, Preziosi P, et al. Oxidative and nitrosative modifications of biliverdin reductase-A in the brain of subjects with Alzheimer's disease and amnestic mild cognitive impairment. J Alzheimers Dis. 2011;25(4):623-33.
- Barone E, Di Domenico F, Cenini G, Sultana R, Cini C, Preziosi P, et al. Biliverdin reductase--a protein levels and activity in the brains of subjects with Alzheimer disease and mild cognitive impairment. Biochim Biophys Acta. 2011;1812(4):480-7.
- Weidinger A, Milivojev N, Hosmann A, Duvigneau JC, Szabo C, Törö G, et al. Oxoglutarate dehydrogenase complex controls glutamate-mediated neuronal death. Redox Biol. 2023;62:102669.
- Maines MD. The heme oxygenase system: a regulator of second messenger gases. Annu Rev Pharmacol Toxicol. 1997;37:517-54.
- Dwyer BE, Nishimura RN, Lu SY. Differential expression of heme oxygenase-1 in cultured cortical neurons and astrocytes determined by the aid of a new heme oxygenase antibody. Response to oxidative stress. Brain Res Mol Brain Res. 1995;30(1):37-47.
- Ahmad AS, Zhuang H, Doré S. Heme oxygenase-1 protects brain from acute excitotoxicity. Neuroscience. 2006;141(4):1703-8.
- Arai-Gaun S, Katai N, Kikuchi T, Kurokawa T, Ohta K, Yoshimura N. Heme oxygenase-1 induced in muller cells plays a protective role in retinal ischemia-reperfusion injury in rats. Invest Ophthalmol Vis Sci. 2004;45(11):4226-32.
- Baranano DE, Rao M, Ferris CD, Snyder SH. Biliverdin reductase: a major physiologic cytoprotectant. Proc Natl Acad Sci U S A. 2002;99(25):16093-8.
- Doré S, Sampei K, Goto S, Alkayed NJ, Guastella D, Blackshaw S, et al. Heme oxygenase-2 is neuroprotective in cerebral ischemia. Mol Med. 1999;5(10):656-63.
- Uittenbogaard M, Chiaramello A. Mitochondrial biogenesis: a therapeutic target for neurodevelopmental disorders and neurodegenerative diseases. Curr Pharm Des. 2014;20(35):5574-93.
- Scarpulla RC. Transcriptional activators and coactivators in the nuclear control of mitochondrial function in mammalian cells. Gene. 2002;286(1):81-9
- Iadecola C, Zhang F, Xu S, Casey R, Ross ME. Inducible nitric oxide synthase gene expression in brain following cerebral ischemia. J Cereb Blood Flow Metab. 1995;15(3):378-84.

Round 2
Reviewer 1 Report
Comments and Suggestions for Authors
This reviewer congratulates the Authors for the excellent job done in this round of revision.